# Medical Management of Pulmonary Arterial Hypertension: Current Approaches and Investigational Drugs

**DOI:** 10.3390/pharmaceutics15061579

**Published:** 2023-05-24

**Authors:** Qi Jin, Dandan Chen, Xiaochun Zhang, Feng Zhang, Dongxiang Zhong, Dawei Lin, Lihua Guan, Wenzhi Pan, Daxin Zhou, Junbo Ge

**Affiliations:** 1Department of Cardiology, Zhongshan Hospital, Fudan University, Shanghai Institute of Cardiovascular Diseases, 180 Fenglin Road, Xuhui District, Shanghai 200032, China; drjinqi@foxmail.com (Q.J.);; 2National Clinical Research Center for Interventional Medicine, 180 Fenglin Road, Xuhui District, Shanghai 200032, China; 3Department of Cardiology, Jinshan Hospital, Fudan University, 1508 Longhang Road, Shanghai 201508, China; 4Department of Cardiology, Shanghai East Hospital, Shanghai Tongji University School of Medicine, 150 Jimo Road, Shanghai 200120, China

**Keywords:** pulmonary arterial hypertension, medical management, endothelin pathway, nitric oxide pathway, prostacyclin pathway, targeted agents, new drugs, prognosis

## Abstract

Pulmonary arterial hypertension (PAH) is a malignant pulmonary vascular syndrome characterized by a progressive increase in pulmonary vascular resistance and pulmonary arterial pressure, which eventually leads to right heart failure and even death. Although the exact mechanism of PAH is not fully understood, pulmonary vasoconstriction, vascular remodeling, immune and inflammatory responses, and thrombosis are thought to be involved in the development and progression of PAH. In the era of non-targeted agents, PAH had a very dismal prognosis with a median survival time of only 2.8 years. With the deep understanding of the pathophysiological mechanism of PAH as well as advances in drug research, PAH-specific therapeutic drugs have developed rapidly in the past 30 years, but they primarily focus on the three classical signaling pathways, namely the endothelin pathway, nitric oxide pathway, and prostacyclin pathway. These drugs dramatically improved pulmonary hemodynamics, cardiac function, exercise tolerance, quality of life, and prognosis in PAH patients, but could only reduce pulmonary arterial pressure and right ventricular afterload to a limited extent. Current targeted agents delay the progression of PAH but cannot fundamentally reverse pulmonary vascular remodeling. Through unremitting efforts, new therapeutic drugs such as sotatercept have emerged, injecting new vitality into this field. This review comprehensively summarizes the general treatments for PAH, including inotropes and vasopressors, diuretics, anticoagulants, general vasodilators, and anemia management. Additionally, this review elaborates the pharmacological properties and recent research progress of twelve specific drugs targeting three classical signaling pathways, as well as dual-, sequential triple-, and initial triple-therapy strategies based on the aforementioned targeted agents. More crucially, the search for novel therapeutic targets for PAH has never stopped, with great progress in recent years, and this review outlines the potential PAH therapeutic agents currently in the exploratory stage to provide new directions for the treatment of PAH and improve the long-term prognosis of PAH patients.

## 1. Introduction

Pulmonary hypertension (PH) refers to an abnormal hemodynamic and pathophysiological state that is not an independent disease itself but includes a group of clinical conditions, which can be derived from lesions of the pulmonary vessels themselves or secondary to other cardiopulmonary or systemic diseases, characterized by pathological pulmonary vascular remodeling and a progressive increase in pulmonary vascular resistance (PVR), ultimately leading to right heart failure or even death. The hemodynamic diagnostic criteria for PH are: rest mean pulmonary artery pressure (mPAP) ≥ 25 mmHg measured by right heart catheterization at sea level [1]. Although the 6th World Symposium on Pulmonary Hypertension (WSPH) and 2022 European Society of Cardiology/European Respiratory Society (ESC/ERS) Guidelines for the diagnosis and treatment of PH recommended modifying the hemodynamic diagnostic criteria for PH from mPAP ≥ 25 mmHg to mPAP > 20 mmHg (mean ± 2 standard deviations) [2,3], this proposal has aroused widespread controversy, and the conclusions of different studies are inconsistent [4,5,6,7,8]; hence, these diagnostic criteria have not been adopted in China at present due to their insufficient basis in relevant studies [9,10]. Clinically, PH is divided into five categories according to the etiology, pathological features, hemodynamic features, and clinical diagnosis and treatment strategies, namely pulmonary arterial hypertension (PAH), PH associated with left heart disease, PH associated with lung diseases and/or hypoxia, PH associated with pulmonary artery obstructions, and PH with unclear and/or multifactorial mechanisms. According to hemodynamic status, PH can be divided into precapillary PH and postcapillary PH (Table 1), and the latter is further divided into isolated post-capillary PH and combined post- and pre-capillary PH [3,11]. The aim of clinical and hemodynamic classification is to better guide the treatment of PH, and different classifications will directly affect the therapeutic strategy of PH.

## 2. PAH Pathogenesis and Pathophysiology

Different subtypes of PH have divergent pathogenesis and pathophysiological features, which are closely related to the diagnosis and treatment. This review mainly focuses on PAH, which is pathologically characterized by small arteriolar intimal hyperplasia; medial hypertrophy; and adventitial fibrosis with varying degrees of inflammatory responses, minimal fibrinoid necrosis, and plexiform lesions that may develop in the late stages. At present, the pathogenesis of PAH is not fully understood, and abnormal pulmonary vasoconstriction and pulmonary vascular remodeling are two important pathophysiological processes of PAH, while previous studies have suggested that genetic and epigenetic modifications, immune and inflammatory disorders, estrogen dysfunction, oxidative-stress-related signaling pathways, and ion channel abnormalities are involved in the occurrence and development of PAH [12,13].

Normal pulmonary vascular beds have strong diastolic and systolic reserve capacity to accommodate the need for increased pulmonary blood flow. However, this ability is compromised in the pulmonary vascular bed among patients with PAH, resulting in increased pulmonary arterial pressure and resistance at rest, especially during exercise. Initially, the right ventricle compensates for the increase in afterload by increasing contractility and ventricular wall thickness, and the right ventricle can maintain normal cardiac output at rest, but cardiac output is somewhat impaired during exercise. As the disease progresses, the right ventricular–pulmonary artery uncouples and cardiac output begins to decline at rest, and as the right heart function deteriorates further, the right ventricle dilates and fails, while the compression of the enlarged right ventricle on the left ventricle causes left ventricular dysfunction [14].

## 3. PAH Risk Stratification

Precise risk stratification is the basis of PAH treatment, and the development of reasonable and personalized treatment strategies according to different risk levels is critical to improving the long-term prognosis of patients and lowering patient mortality. At present, there is no single indicator that can accurately determine the severity of the disease and evaluate the prognosis of patients, so it is necessary to synthesize multiple clinical indicators for comprehensive evaluation.

In the 2015 ESC/ERS guidelines for the diagnosis and treatment of PH, PAH patients were classified into low-risk, intermediate-risk, and high-risk groups based on clinical manifestations, symptom progression, syncopal attack, World Health Organization functional class (WHO-FC), 6 minute walk distance (6MWD), cardiopulmonary exercise test, plasma brain natriuretic peptide (BNP) or N-terminal (NT)-pro BNP (NT-proBNP) level, imaging examination, and hemodynamic characteristics of right heart failure, and their 1-year mortality rates were estimated to be <5%, 5–10%, and >10%, respectively [1]. However, the aforementioned risk assessment model contains many indicators, some technical procedures are not carried out in all hospitals, and certain parameters are not indicators for routine review during follow-up, making it inconvenient for clinicians, especially doctors in primary hospitals, to operate and implement them. In 2018, the 6th WSPH released a simplified version based on the original risk stratification model, including cardiac function, exercise tolerance, blood biomarkers, echocardiography, and hemodynamic parameters, with fewer indicators and a greater ease of implementation, as well as a detailed definition of low-, intermediate-, and high-risk criteria, making the risk stratification clearer and more convenient for clinical application [15]. It should be emphasized that risk stratification assessment should not be limited to the baseline state, but should be dynamic and accompany the entire treatment course of the patient. At present, the recommended risk stratification assessment scales for PAH mainly include: the French Pulmonary Hypertension Network Risk Assessment Equation [16], REVEAL risk score [17], Pulmonary Hypertension Connection Equation [18], 2015 ESC/ERS PH guideline risk assessment model [1], COMPERA ESC/ERS abbreviated risk score assessment [19], Scottish composite score [20], and COMPERA 2.0 [21]. However, the above risk stratification scales are only applicable to adult PAH patients, and there is still a lack of unified risk stratification scales for other types of PH and pediatric PH. Although some studies suggest that the above scales are also applicable to patients with chronic thromboembolic pulmonary hypertension (CTEPH), they have not been widely recognized [22,23].

## 4. Pharmacological Therapy for PAH

Pharmacological therapy for PAH includes general and PAH targeted agent therapy, the former including inotropes and vasopressors, diuretics, anticoagulants, and vasodilators. For patients with confirmed PAH in specialist centers, general medical measures and supportive care are administered. Acute vasoreactivity testing (AVT) should be performed in patients with idiopathic PAH (IPAH), heritable PAH (HPAH), and drug/toxin-associated PAH; if positive, these patients can largely benefit from calcium channel blockers (CCBs). If negative, it is recommended that low-risk or intermediate-risk patients receive initial monotherapy or oral combination therapy, with initial combination therapy for high-risk patients. If the drug treatment is not effective, sequential dual or triple therapy may be considered, and if that fails, lung transplantation may be considered (Figure 1) [1,15].

Low-risk individuals, such as those who are elderly, in poor family economic situations, or have a stable disease and are on long-term drug therapy, should be treated with monotherapy, and if the patient remains low-risk after 3–6 months of treatment, the original regimen can be maintained, and regular follow-up continues. Initial oral combination therapy is employed in intermediate-risk patients. If the patient remains intermediate-risk at follow-up, intensive treatment including raising the dose of drugs and expanding the type of combined drugs should be utilized so as to downgrade to low-risk status. For high-risk PAH patients, initial combination therapy including intravenous or subcutaneous prostacyclin drugs is recommended, and aggressive therapy such as lung transplantation should be adopted if the patient remains high-risk at follow-up (Figure 1) [15].

### 4.1. Inotropes and Vasopressors

In the presence of right ventricular failure manifesting with persistent hypotension, inotropes and/or vasopressors are often indicated in all forms of PH to ensure hemodynamic stability and peripheral perfusion. Digoxin enhances cardiac contractility and improves cardiac output and can be used to lower the ventricular rate in PAH patients who develop supraventricular tachycardia. However, a recent retrospective study revealed that chronic digoxin use was associated with higher all-cause mortality and heart failure hospitalization during a median 2.1-year follow-up period [24], though further randomized studies are needed to validate these results. Because digoxin is principally cleared by the kidneys, plasma concentrations should be closely monitored to avoid toxicity, especially in patients with impaired renal function. The monitoring of blood electrolytes for patients treated with diuretics is essential because hypomagnesemia and hypokalemia augment the inhibitory effect of digoxin on the sodium–potassium pump, resulting in proarrhythmic consequences [25]. Levosimendan is a calcium sensitizer with inotropic, pulmonary vasodilatory, and cardioprotective features that has been shown to be an effective and safe treatment strategy for PAH and PH associated with left heart disease [26]. Levosimendan treatment improved cardiac output, decreased right ventricular afterload, and alleviated pulmonary vascular remodeling in SU5416/hypoxia-induced PAH rat models [27]. The most recent systematic review and meta-analysis demonstrated that levosimendan improved right-ventricular systolic function and reduced pulmonary arterial pressure in patients with cardiac dysfunction [28]. For PAH associated with connective tissue disease (CTD-PAH) patients, levosimendan was effective in improving acute decompensated right heart failure, systemic hemodynamics, and in-hospital survival, but no medium- or long-term survival benefit was observed [29]. For patients with low cardiac output, dobutamine and milrinone are the most extensively used inotropes, but levosimendan appears to be more efficacious than dobutamine in animal models of right heart failure, though convincing clinical evidence is still lacking [30]. Patients with systemic hypotension may require additional vasopressors; norepinephrine and vasopressin are the drugs of choice, though vasopressin may be more advantageous because of its pulmonary vasodilator effect [30].

### 4.2. Diuretics

Fluid retention, increased central venous pressure, hepatic congestion, and polyserous effusion might be observed in PAH patients who proceed to decompensated right heart failure. Gastrointestinal edema can lead to digestive problems and malabsorption, interfere with the absorption and utilization of targeted agents, and thus affect their efficacy [30]. Currently, no randomized controlled study has been conducted on the application of diuretics in PAH patients. Nevertheless, clinical experience suggests that diuretics can effectively reduce edema and right heart failure symptoms when used concurrently with PAH targeted drugs. The choice of drug type and dose is best determined by PAH specialists. Loop diuretics (furosemide, torasemide) and aldosterone receptor antagonists (spironolactone) are commonly used, though the latter should be taken with caution in individuals at risk of renal failure, diabetes, or hyperkalemia or those receiving drugs with potential nephrotoxicity [25]. Recent studies have suggested that tolvaptan, a vasopressin V2 receptor antagonist, can effectively increase urine volume, reduce body weight, and improve clinical symptoms and signs without reducing the effective circulating blood volume, but its exact efficacy and safety in PAH patients with right heart failure still need to be confirmed by further clinical studies [31,32]. To minimize hypovolemia, renal insufficiency and electrolyte imbalance, body weight and blood biochemical indicators such as renal function and blood electrolytes must be monitored throughout diuretic therapy.

### 4.3. Anticoagulants

Micro-thrombosis is one of the pathophysiological features of PAH. Previous autopsy studies have found that PAH patients have a higher incidence of in situ thrombosis in pulmonary arterioles [33], which may be the result of endothelial cell dysfunction, platelet activation, coagulation, and fibrinolysis imbalance [25]. A meta-analysis showed that anticoagulant therapy with warfarin improved the prognosis of IPAH patients, but the included studies were mainly retrospective studies before the era of targeted therapy and lacked randomized controlled study data, which were not consistent with the results of large prospective PAH registries in recent years. The 2015 ESC/ERS Guidelines for the Diagnosis and Treatment of PH recommend that long-term anticoagulant therapy may be considered for IPAH, HPAH, and PAH due to weight-loss drugs if no contraindications exist, while there is insufficient evidence to suggest that patients with other types of PAH benefit from anticoagulant therapy [1]; anticoagulant therapy does not improve survival and may even increase the risk of death in patients with systemic sclerosis-associated PAH; it is therefore not recommended for such patients [34]. It is clinically necessary to conduct a comprehensive analysis regarding the etiology and status of PAH patients to individualize the risks and benefits of anticoagulant therapy [15]. Anticoagulation is recommended for PAH patients with thrombosis or high-risk factors for thrombosis, and patient compliance and bleeding risk should be considered. Patients with frequent hemoptysis; a high risk of bleeding; and the inability to regularly detect anticoagulant parameters, such as portopulmonary hypertension, Eisenmenger’s syndrome with hemoptysis, hereditary hemorrhagic telangiectasia, and pulmonary capillary hemangiomatosis, should not receive anticoagulation [35]. Once diagnosed, lifelong anticoagulation therapy is recommended in patients with CTEPH if there is no contraindication to anticoagulation, and CTEPH patients receiving new oral anticoagulants present a similar or lower incidence of major bleeding compared with vitamin K antagonists but an increased risk of recurrent venous thromboembolism [36,37]. However, among PAH patients receiving anticoagulation, the Pulmonary Hypertension Association Registry disclosed that anticoagulation was not associated with higher mortality but was associated with a poorer quality of life and increased emergency department visits, hospitalizations, and hospital days [38]. At present, most studies use warfarin as an anticoagulant, and there is still a paucity of relevant data on the effect of new oral anticoagulant on the prognosis of PAH patients.

### 4.4. General Vasodilators

AVT is a fast-acting pharmacological test performed with selective vasodilator agents (e.g., iloprost, adenosine, and nitric oxide) during right heart catheterization to monitor changes in hemodynamics in PAH patients to predict whether patients respond to CCB therapy. Favorable criteria include a drop in mPAP of ≥10 mmHg, a decrease in absolute mPAP to ≤40 mmHg, and unchanged or increased cardiac output. Positive individuals with IPAH, HPAH, and drug-related PAH are recommended to be treated with adequate doses of CCBs, but negative patients may not benefit from CCBs and may even experience a worsening in their condition because of possible serious side effects such as hypotension, syncope, and right heart failure [1]. AVT is positive in roughly 10% of IPAH patients, and only about half of positive patients have a persistent response. Other treatments should be utilized instead If AVT turns negative. Therefore, patients should be constantly followed up, and a comprehensive evaluation should be performed again after 3–4 months. Nifedipine and amlodipine are considered in patients with slow heart rates, and diltiazem tends to be used in patients with rapid heart rates. It is recommended to start at a low dose and gradually increase to the highest tolerated dose. For other types of PAH patients, AVT does not predict the long-term efficacy of CCBs and is therefore not recommended. Angiotensin-converting enzyme inhibitors, angiotensin II receptor antagonists, β-blockers, nitrates, and ivabradine are not recommended for the treatment of PAH. If the above drugs are required due to left heart disease (hypertension, coronary heart disease, etc.), blood pressure and heart rate should be closely monitored, and drug–drug interactions should be noted [39].

### 4.5. Treatment of Anemia

Iron deficiency is common in PAH patients and is frequently associated with severity and prognosis in patients with IPAH, PAH associated with congenital heart disease, and CTD-PAH [40,41,42]. Oral iron supplementation has a poor absorption effect, and intravenous iron supplementation is preferred. Retrospective studies showed that intravenous iron therapy could safely correct iron deficiency status; improve 6MWD, WHO-FC, and risk stratification; and reduce rehospitalization rates in PAH patients [43]. Recent randomized, double-blind, placebo-controlled studies have shown that intravenous iron is well-tolerated, but no benefit in exercise tolerance, hemodynamic parameters, and cardiac function has been observed over 12 weeks [44]. Further studies are required to confirm the safety and efficacy of iron supplementation.

### 4.6. PAH Targeted Therapy

Treatment strategies vary across PH types, and the PAH targeted therapy discussed herein is not applicable to patients with other types of PH, especially those with Group 2 or 3 PH, among whom targeted drug therapy remains widely controversial. Currently, the clinically employed targeted agents mainly include endothelin receptor antagonists (ERAs), phosphodiesterase type 5 (PDE5) inhibitors, soluble guanylate cyclase agonists, prostacyclin analogs, and prostacyclin receptor agonists (Figure 2, Table 2). The diverse targets of targeted medications provide a theoretical basis for combination therapy.

#### 4.6.1. Monotherapy

##### Endothelin Pathway

Endothelin-1 is a potent endogenous vasoconstrictor that is overexpressed in the pulmonary vasculature of PAH patients and can cause pulmonary vasoconstriction, promote cell mitosis, and contribute to the development and progression of PAH by binding to endothelin receptors A and B in pulmonary vascular smooth muscle cells. ERAs block the activity of endothelin by targeting endothelin receptors A and/or B. Bosentan, ambrisentan, and macitentan are the only clinically approved ERAs for PAH. Ambrisentan is a selective endothelin receptor A antagonist, while the others are nonselective dual ERAs. The main side effects of ERAs are hepatotoxicity and peripheral edema, and strict contraception is required when using such medications due to the potential teratogenic effects of ERAs.

Bosentan, a dual endothelin receptor A and B antagonist, was the first oral targeted agent approved for the treatment of PAH. Bosentan was approved by US Food and Drug Administration (FDA) for marketing in December 2001, introduced into the Chinese market in June 2006, and approved by China’s FDA for the treatment of paediatric PAH patients in September 2019. Bosentan, together with selexipag, riociguat, and macitentan, entered the Chinese national health insurance drug list in November 2019. The BREATHE-1 study showed that Bosentan could improve PVR and exercise tolerance and postpone the time to clinical deterioration [45]. Bosentan starts at 62.5 mg bid for 4 weeks and then increases to a maintenance dose of 125 mg bid. Since Bosentan is metabolized by the liver and may cause transaminase elevations in a dose-dependent manner, with the recovery of liver function following dose reduction or discontinuation, liver function should be monitored regularly during treatment.

Ambrisentan predominantly affects endothelin receptor A and is the least hepatotoxic ERA. It was approved by the US FDA for PAH in June 2007. Results from randomized placebo-controlled studies of ARIES-1, ARIES-2, and ARIES-E consistently showed that ambrisentan improved exercise tolerance, WHO-FC, pulmonary vascular hemodynamics, and quality of life and delayed disease progression and clinical worsening in PAH patients [46,47]. In adult patients, ambrisentan is initiated at 5 mg qd (administered orally in the fasted or fed state); adjustment to 10 mg qd can be considered if tolerated. Ambrisentan is contraindicated in patients with idiopathic pulmonary fibrosis with or without secondary PH [48].

Macitentan is a novel oral dual ERA with greater tissue penetration and receptor affinity than bosentan and ambrisentan. In October 2013, macitentan was approved by the US FDA for marketing and in China in October 2017 at a dose of 10 mg qd. The SERAPHIN study showed that macitentan reduced the risk of PAH-related events or all-cause mortality by 45% and significantly improved WHO-FC and 6MWD compared with placebo [49]. The serious adverse reaction of macitentan is anemia, which necessitates the intensive routine monitoring of blood rather than liver function.

##### Nitric Oxide Pathway

Nitric oxide (NO) produced by endothelial cells is an important vasodilator that binds to soluble guanylate cyclase and mediates the synthesis of cyclic guanosine monophosphate (cGMP), which subsequently activates cGMP-dependent protein kinases, resulting in the opening of potassium channels and a decrease in intracellular calcium concentrations, thereby dilating blood vessels. PDE5, a cGMP-degrading enzyme, is abundant in pulmonary vessels, and its inhibitors can reduce cGMP degradation via the NO/cGMP pathway, raise its concentration to relax blood vessels, and inhibit smooth muscle cell proliferation. Current agents that increase cGMP include PDE5 inhibitors (sildenafil, tadalafil, and vardenafil) and soluble guanylate cyclase stimulators (riociguat).

Sildenafil was the first approved PDE5 inhibitor for the treatment of PAH. Early in March 1998, Sildenafil was approved by the US FDA for the treatment of male erectile dysfunction, then entered the Chinese market in 2000, and was approved for the treatment of PAH in the US and Europe in 2005; however, it was not approved for the treatment of adult PAH in China until February 2020. The SUPER-1 study showed that sildenafil significantly improved hemodynamics and 6MWD, with the benefits lasting for over 1 year at follow-up [50]. The SUPER-2 study showed that at the end of 3-year treatment, 46% and 29% of patients had sustained improvements in 6MWD and WHO-FC, respectively, compared to baseline measurements before the SUPER-1 study, with a 3-year survival rate of 79% [51]. Several clinical studies have confirmed that sildenafil can improve the clinical symptoms and cardiac function of PAH patients in China, with good safety and tolerability [52,53]. The recommended dose for adults is 20–80 mg tid. Headache, gastrointestinal discomfort, flushing, and muscle and joint pain are common side effects.

Tadalafil is a slow-acting PDE5 inhibitor that was approved by the US FDA in 2003 for the treatment of erectile dysfunction in men and in May 2009 for the treatment of PAH. The PHIRST-1 study demonstrated that tadalafil significantly improved 6MWD, delayed time to clinical worsening, reduced the incidence of clinical worsening, and improved patients’ health-related quality of life [54]. These improvements in 6MWD were maintained in the PHIRST-2 trial throughout the extended 52-week observation period [55]. Switching between sildenafil and tadalafil is generally well-tolerated, and doses of 20 and 40 mg qd are recommended with the same adverse effects as sildenafil [56].

Vardenafil, a highly selective PDE5 inhibitor, was initially licensed in the European Union in 2003 for the treatment of male erectile dysfunction. It was introduced into China in 2004 and is currently off-label and used for the treatment of PAH. A randomized, double-blind, placebo-controlled study enrolled 66 Chinese PAH patients randomly assigned to vardenafil versus placebo with a primary endpoint of 6MWD. Vardenafil was well-tolerated and significantly improved exercise tolerance, increased the cardiac index, and decreased mPAP and PVR in Chinese PAH patients [57]. A phase-2b, open-label study to evaluate the safety and efficacy of a single dose of inhaled vardenafil in patients with PAH is ongoing (NCT04266197).

Riociguat is the first oral soluble guanylate cyclase agonist to be independent of NO levels and act directly on intracellular soluble guanylate cyclase, boosting cGMP levels, improving sensitivity to low levels of NO, relaxing pulmonary vessels, and presenting antiproliferative and antifibrotic properties. It was approved for the treatment of PAH by the US FDA in October 2013 and for CTEPH by Japan in January 2014. In March 2014, China’s FDA accepted the import application of riociguat tablets. PATENT-1 and PATENT-2 studies included PAH patients with WHO-FC II/III, and riociguat was found to improve clinical symptoms, exercise tolerance, WHO-FC, PVR, and time to clinical worsening [58,59]. In addition, CHEST-1 and CHEST-2 trials confirmed that riociguat significantly improved exercise tolerance, NT-proBNP levels, WHO-FC, and PVR in patients with inoperable or persistent/recurrent CTEPH after pulmonary endarterectomy, with good safety and tolerability [60,61]. Riociguat is the only targeted agent approved so far for the treatment of CTEPH. The PATENT PLUS study evaluated the safety and efficacy of riociguat in combination with sildenafil in PAH patients and found that the incidence of hypotension was significantly higher in the combination group, with no significant differences in hemodynamic parameters or exercise capacity compared with controls, and therefore the combination of riociguat and PDE5 inhibitors is not recommended [62]. The recommended starting dose for adults is 1 mg tid, titrated gradually according to patient tolerance, increasing by 0.5 mg every 2 weeks to a maximum dose of 2.5 mg tid.

##### Prostacyclin Pathway

Prostacyclin is generated by vascular endothelial cells; functions through prostacyclin receptors to promote cyclic adenosine monophosphate production; and inhibits pulmonary artery smooth muscle cell proliferation and platelet aggregation, exerting vasodilator effects. In PAH patients, prostacyclin levels and function are abnormally reduced, and synthetic prostacyclin analogs can be used to treat PAH. Currently approved prostacyclin pathway drugs are divided into: prostacyclin analogs (epoprostenol, iloprost, treprostinil, and beraprost) and prostacyclin receptor agonists (selexipag).

Epoprostenol was the earliest targeted drug used for the treatment of PAH and was marketed in Europe and the United States in 1995, opening the way for PAH targeted therapy. Intravenous epoprostenol significantly improved survival, WHO-FC, exercise tolerance, and hemodynamic parameters in IPAH patients with WHO-FC III-IV [63,64]. It has good efficacy in all subcategories of PAH patients and is currently the preferred therapeutic drug for PAH patients with WHO-FC IV. Unfortunately, this drug has not yet entered the Chinese market [65]. Epoprostenol has a short half-life and requires continuous intravenous pumping to maintain the therapeutic effect, and abrupt withdrawal should be avoided to prevent rebound phenomena. The dose is usually started at 2–4 ng/kg/min and increased slowly, with a general target dose of 20–40 ng/kg/min, while the highest dose can reach more than 100 ng/kg/min. The adverse effects of epoprostenol mainly include delivery system abnormalities, local infections, catheter obstruction, and sepsis; headache, hypotension, nausea, and diarrhea may also occur at high doses.

Iloprost is a prostacyclin analog. In December 2004, the US FDA approved inhaled iloprost for the treatment of IPAH patients with WHO-FC III/IV. In April 2006, China’s FDA authorized its official entry onto the Chinese market, making it the first targeted drug for the treatment of PAH in China. Unfortunately, for various reasons, the drug has gradually been withdrawn from the Chinese market since 2015. It has a short half-life, rapid onset of action, and can be injected intravenously or nebulized, and the inhaled dosage form requires 6–9 doses of 10–20 μg per day. Because inhaled iloprost has a rapid onset of action (2–5 min), it can be used not only for AVT, but also for the rescue of PAH crisis. Intravenous iloprost can be used to treat PAH or CTEPH patients with severe right heart failure, and numerous studies have consistently shown that iloprost can improve exercise tolerance, hemodynamic parameters, and long-term survival in PAH patients, with mild adverse reactions and good tolerance [66,67,68]. Intravenous application requires pumping from the central vein at a dose of 0.5–4 ng/kg/min. Common adverse reactions include facial hot flashes and jaw pain.

Treprostinil is a stable and slow-acting prostacyclin analog developed by United Therapeutics that was approved by the US FDA for the treatment of PAH in May 2002; it can be administered intravenously or subcutaneously, but also in inhaled and oral dosage forms, and the subcutaneous dosage form is stable at room temperature and continuously administered by a micropump. The most common adverse reactions of subcutaneous administration are injection site pain and digestive system symptoms, followed by facial hot flashes and headache. Treprostinil is more stable than epoprostenol when administered intravenously but carries a risk of central venous infection. Several studies have shown that intravenous and subcutaneous treprostinil dose-dependently increases 6MWD and improves patient symptoms, quality of life, and hemodynamic status [69,70,71,72]. Switching from continuous intravenous epoprostenol to intravenous or subcutaneous treprostenol has also been shown to be safe and effective [73]. The INCREASE study found that inhaled treprostinil significantly improved cardiac function, exercise tolerance, pulmonary function, and poor prognosis in patients with PH associated with interstitial lung disease [74,75], and patients who continued to receive inhaled treprostinil experienced fewer disease progression events after an initial clinical worsening event compared with patients who received the placebo [76], and based on the above studies, inhaled treprostinil was approved by the US FDA for the treatment of PH associated with interstitial lung disease in May 2022. A phase-3, randomized, placebo-controlled, double-blind study to evaluate the safety and efficacy of inhaled treprostinil in patients with PH due to chronic obstructive pulmonary disease (PERFECT study, NCT03496623) was terminated following a routine safety and efficacy analysis, and its results will soon be released. Another open-label study of inhaled treprostinil in sarcoidosis-related PH (SAPPHIRE study, NCT03814317) was expected to be completed in October 2022.

Beraprost, the first chemically stable oral prostacyclin analog, is currently approved for the treatment of PAH in only Japan and South Korea. European and American studies have shown that Beraprost treatment for 3–6 months can improve exercise tolerance in PAH patients with WHO-FC II/III, but no long-term benefit has been observed. Drug-related adverse reactions are common, such as headache, jaw pain, and flushing [77,78].

Selexipag is a slow-acting oral selective prostacyclin IP receptor agonist that causes pulmonary vasodilatation, and its active metabolite is highly selective for the human prostacyclin receptor, so its adverse reactions are greatly reduced compared with previous prostacyclin analogs. In December 2015, the US FDA approved selexipag for the treatment of adult PAH. The GRIPHON study showed that selexipag reduced PVR, significantly increased the cardiac index, and was well-tolerated in PAH patients [79], and subgroup analysis suggested that selexipag lowered the risk of composite morbidity/mortality events by 41% in CTD-PAH patients [80]. Selexipag was approved in China in December 2018, and a recent study demonstrated that triple combination therapy with selexipag effectively improved exercise capacity, right heart function, risk assessment, and prognosis among Chinese PAH patients with acceptable tolerability. Additionally, studies have shown that selexipag significantly improves PVR and other hemodynamic parameters in patients with CTEPH, but there is no significant improvement in 6MWD and WHO-FC [81], and further studies are needed to confirm the role of selexipag in CTEPH, while studies on the safety and efficacy of oral selexipag in patients with sarcoidosis-associated PH (SPHINX study, NCT03942211) and schistosomiasis-associated PH (SELSCH study, NCT04589390) are ongoing. The current recommended dose for adults is 200 µg bid, increased weekly by 200 µg up to a maximum of 1600 µg bid. Adverse reactions are consistent with other prostacyclin drugs, such as headache, diarrhea, nausea, flushing, myalgia, and jaw pain.

#### 4.6.2. Combination Therapy

Combination therapy is recommended for intermediate- and high-risk patients except for a few special circumstances requiring initial monotherapy. This refers to the combined use of two or more targeted drugs with different signaling pathways, with the purpose of maximizing drug efficacy, reducing drug toxicity, lowering drug dose, and changing ineffective or decreased-efficacy treatments into effective treatments. Although great progress has been made in PAH drug therapy in recent years, the long-term prognosis of PAH patients remains unsatisfactory. The pathogenesis of PAH is complex, and combination therapy with drugs targeting different signaling pathways is theoretically more effective than single-drug therapy. PAH combination therapy is divided into sequential combination therapy and initial combination therapy. Sequential combination therapy refers to the application of a certain drug first, followed by other targeted drug therapy when the effect is not good; the latter refers to combination therapy using two or more drugs at the initial start of treatment. In recent years, the results of multiple randomized controlled trials have shown that both sequential combination therapy and initial combination therapy can significantly reduce the occurrence of clinical worsening in PAH patients.

##### Dual Combination Therapy


1.ERAs Combined with PDE5 Inhibitors


ERAs combined with PDE5 inhibitors are currently a widely used combination regimen in clinical practice; both are oral preparations and are easy to take, and their safety and efficacy have been fully verified. The COMPASS-1 study was a prospective, multicenter study that included stable PAH patients, all of whom received bosentan 125 mg bid for more than 12 weeks before enrollment, and changes in pulmonary hemodynamics were assessed 60 min after the provisional administration of sildenafil 25 mg. PVR was found to be considerably reduced, suggesting that sildenafil combined with bosentan could produce immediate and favorable pharmacological benefits without serious adverse events [82]. Real-world studies revealed that after 3–4 months of bosentan combined with sildenafil treatment in PAH patients, WHO-FC, exercise tolerance and hemodynamic parameters improved significantly, as did the 1-, 3- and 5-year survival rates [83]. A subgroup analysis of the SERAPHIN study showed that sequential treatment with macitentan significantly reduced composite endpoint events in the setting of backgroud therapy, with approximately 60% of patients taking PDE5 inhibitors as background medications, primarily sildenafil [49]. The AMBITION study showed that an initial combination of ambrisentan and tadalafil reduced by 50% the risk of primary endpoint events and significantly reduced PAH rehospitalization compared with monotherapy. Initial combination therapy significantly lowered NT-proBNP levels and improved 6MWD compared with monotherapy during 24-week treatment course [84]. Four hundred and five PAH patients with WHO-FC II/III were included in the PHIRST study, and about half of the patients were treated with tadalafil in addition to bosentan. Patients in the 40 mg tadalafil group had significantly higher 6MWD, delayed clinical deterioration, a reduced clinical deterioration rate, and better health-related quality of life [55].
2.ERAs Combined with Prostacyclin

The BREATHE-2 study included 33 IPAH and CTD-PAH patients with WHO-FC III/IV who received 2 days of epoprostenol followed by randomization to bosentan and placebo groups for 16 weeks, and hemodynamic parameters, exercise tolerance, and WHO-FC tended to improve in the combined treatment group compared with the control group (*p* > 0.05) [85]. The STEP study found that PAH patients with WHO-FC III/V (55% IPAH, 45% associated PAH) treated with sequential combination therapy of bosentan and inhaled iloprost significantly increased 6MWD and improved WHO-FC, hemodynamics, and time to clinical deterioration [86]. The COMBI study assessed the efficacy and safety of combined inhaled iloprost in stable WHO-FC III IPAH patients who had been treated with bosentan 125 mg bid orally for 12 weeks; three patients in the experimental group experienced clinical deterioration at week 12, while no significant difference was observed in each observation endpoint, and the study was finally terminated prematurely [87]. When compared with monotherapy, initial combination therapy with bosentan and inhaled iloprost significantly improved 6MWD, hemodynamics, and quality of life in PAH patients with WHO-FC III/IV [88]. The TRIUMPH I study included 235 PAH patients who worsened despite bosentan or sildenafil therapy and were randomized to the addition of inhaled treprostinil or placebo for 12 weeks; it found significant improvements in exercise tolerance and quality of life in the treprostinil group, but no differences in time to clinical worsening, dyspnea score, and WHO-FC [89]. The FREEDOM-C study suggested that adding oral treprostinil to PAH patients taking ERAs and/or PDE5 inhibitors did not improve 6MWD at 16 weeks [90].
3.PDE5 Inhibitors Combined with Prostacyclin

The PACES study was designed to assess the efficacy and safety of adding oral sildenafil to chronic intravenous epoprostenol in PAH patients and found that sequential epoprostenol combined with sildenafil significantly improved 6MWD, hemodynamic parameters, time to clinical worsening, and quality of life, but increased the incidence of headache and dyspepsia [91]. The addition of sildenafil to inhaled iloprost significantly improved exercise capacity, WHO-FC, and hemodynamic parameters in PAH patients [92,93]. Preliminary studies showed that after administering sildenafil to eight PAH patients who had been receiving subcutaneous treprostinil for a long time and had a stable status for 12 weeks, WHO-FC decreased from III to II in three patients, mean treadmill time increased significantly, and symptoms improved in all patients [94].

##### Triple Combination Therapy

Dual therapy has significant advantages over monotherapy in improving symptoms, reducing clinical events, and prolonging survival, but real-world studies have shown that most patients remain at intermediate-risk levels in the dual-therapy setting, and mPAP and PVR remain well above normal values, suggesting that dual therapy may not be sufficient to control most patients at low-risk levels [83]. The post hoc analysis of the AMBITION study showed that 72% of CTD-PAH patients remained at intermediate and high risk after 4 months of initial dual therapy with ambrisentan and tadalafil, suggesting that most patients remained substandard under initial dual therapy. Multicenter retrospective studies showed that although the initial combination of ambrisentan and tadalafil significantly reduced PVR (11.5 ± 6.5 vs. 7.2 ± 4.1 Wood units), 57% of PAH patients receiving combination therapy did not achieve a low-risk status and required additional treatment [95]. Theoretically, early treatment with triple targeted agents may allow patients to reach a low-risk state earlier and bring survival benefits, but this requires more clinical evidence. Aggressive afterload reduction with initial oral combination therapy and early parenteral prostaglandin therapy may significantly improve the clinical outcomes of PAH patients [96].
1.Sequential Triple Therapy

A retrospective study to evaluate the effect of sequential treatment with intravenous treprostinil on risk stratification and transplant-free survival included 126 PAH patients (98% were intermediate and high-risk patients, 98% had a background of targeted drug combination at baseline, and the proportion of dual therapy reached 78%). During a follow-up of 6–12 months, the proportion of low-risk patients increased from 2% to 19%, and the 5-year transplant-free survival rate reached 94%, suggesting that sequential triple targeted drug therapy can control disease progression and improve patient prognosis [97]. Another retrospective cohort study included 28 patients with Eisenmenger’s syndrome who were treated with PDE5 inhibitors plus ERAs at baseline. After 27 months of parenteral prostacyclin therapy, 6MWD, NT-proBNP, PVR, and cardiac index were significantly improved, with 1-year and 2-year survival rates of 92% and 80%, respectively. No patient discontinued the drug due to side effects during the follow-up period, suggesting that the addition of parenteral prostaglandins to dual agents is safe and effective [98]. In the GRIPHON study, 32.5% of patients were receiving dual ERAs and PDE5 inhibitors at baseline, and a post hoc subgroup analysis showed that selexipag reduced the risk of the primary composite endpoint by 37% in patients with dual therapy, implying that the sequential combination of selexipag with ERAs and PDE5 inhibitors improves long-term outcomes [79].
2.Initial Triple Therapy

An exploratory study included 19 PAH patients with WHO-FC III/V who were initially treated with iloprost, bosentan, and sildenafil, and at 4-month follow-up, WHO-FC, 6MWD, and hemodynamic parameters were significantly improved; at an average follow-up of 32 months, all patients achieved WHO-FC I/II except one patient who underwent heart-lung transplantation at 3 months, and the 3-year survival rate reached 100%, which was much higher than the 49% survival rate expected by the French equation [99]. A previous study assessed the effect of initial triple therapy with ambrisentan, tadalafil, and subcutaneous treprostinil on risk stratification and right heart function in patients with newly diagnosed high-risk IPAH, with significant improvements in WHO-FC, 6MWD, NT-proBNP, hemodynamics, and right heart function at a median follow-up of 2 years, and a reduction in REVEAL risk score by the end of follow-up; 81% of patients were at low risk, 19% at intermediate risk, and all patients survived [100]. Initial triple oral therapy with macitentan, riociguat, and selexipag was well-tolerated and fully improved cardiac function and exercise tolerance in PAH patients [101]. The TRITON study was a prospective randomized controlled study to compare the efficacy and safety of initial triple and dual therapy in newly diagnosed PAH patients and found that initial triple therapy did not significantly improve the primary endpoint PVR compared with traditional dual therapy. Hemodynamic parameters, NT-proBNP levels, and 6MWD were significantly improved in both groups, but there was no statistical difference between the two groups. Of the initial triple therapy patients, 42.9%, and of the initial dual therapy patients, 31.5% had at least one serious adverse reaction event [102]. The latest French retrospective study, which included 1611 PAH patients, analyzed the relationship between initial treatment strategy and survival, and demonstrated that initial triple therapy strategy including parenteral prostacyclins improved the prognosis of high-risk PAH patients; in addition, this study showed for the first time that initial triple therapy including parenteral prostacyclins resulted in a significant survival benefit for intermediate-risk PAH patients compared with dual or single therapy [103].

## 5. Potential Drugs Being Investigated

There are many kinds of clinical study drugs targeting relevant signal targets, which are mainly divided into totally new drugs, drugs with existing drug structure modifications or target integration, and drugs with indications for other conditions. In recent years, rapid progress has been made in the pharmacological treatment of PAH. The following table summarizes the clinical trials regarding PAH drug therapy, and some of these studies have achieved satisfactory preliminary results, providing promising therapeutic directions for the treatment of PAH (Table 3) [104,105].

Due to limited space, only a few potential drugs are introduced here. Numerous studies have demonstrated that imbalances in bone morphogenetic protein antiproliferative signaling and activin/growth differentiation factor pro-proliferative signaling are critical drivers in the development and progression of familial, idiopathic, and acquired PAH. Sotatercept (ACE-011) is a novel fusion protein targeting the TGF-β signaling pathway that binds to activin and growth differentiation factors and restores the balance between pro-proliferative and anti-proliferative signaling pathways, which may reverse pathological vascular remodeling and restore vascular homeostasis. Sotatercept was effective in preventing disease progression in animal models of PAH and may treat PAH by reversing pathological vascular remodeling [106]. The PULSAR study (NCT03496207) was a 24-week phase-2, randomized, double-blind, placebo-controlled study designed to compare the efficacy and safety of sotatercept versus placebo in the treatment of PAH that enrolled 106 adult PAH patients receiving background therapy with the primary endpoint of change from baseline to week 24 in PVR and the secondary endpoint of improvement in 6MWD. The study found an 18% reduction in PVR from baseline and significant improvements in 6MWD and NT-proBNP in patients receiving sotatercept compared with placebo [107]. Sotatercept has revolutionized the current PAH treatment paradigm using three classical signaling pathways, and based on above results, the US FDA granted Breakthrough Therapy Designation to sotatercept for the treatment of PAH patients in April 2020. The PULSAR open-label extension study supported the longer-term safety and durability of the clinical benefits of sotatercept for PAH [108]. The STELLAR study (NCT04576988), a multicenter, double-blind, phase-3 trial, demonstrated that subcutaneous sotatercept every 3 weeks substantially improved exercise capacity, PVR, NT-proBNP, time to death or clinical worsening, and French risk score during a 24-week period among WHO-FC II-III PAH patients on stable background therapy [109]. The SPECTRA trial (NCT03738150) is a phase-2a, single-arm, open-label, multicenter exploratory study to explore the effects of sotatercept plus standard of care in adult PAH patients in WHO-FC III; it is currently completed, and results are forthcoming. A phase-3, randomized, double-blind, placebo-controlled study (HYPERION, NCT04811092) is ongoing to evaluate the efficacy of sotatercept when added to background PAH therapy in newly diagnosed intermediate- or high-risk PAH.

Ralinepag is a new generation of selective, slow-acting, oral prostacyclin IP receptor agonists, which have important effects such as vasodilation, the inhibition of vascular smooth muscle cell proliferation, and the inhibition of platelet aggregation [110]. A 22-week phase-2 randomized, placebo, parallel-group study assessed the efficacy and safety of ralinepag in the treatment of patients with symptomatic PAH and found that ralinepag significantly improved PVR compared with placebo, with good safety, tolerability, and adverse reactions consistent with other prostacyclin drugs; it is expected to be a novel and potent prostacyclin IP receptor agonist for the treatment of PAH [111,112].

TPN171H is a novel selective and orally active PDE5 inhibitor and could significantly reduce right ventricular systolic pressure and hypertrophy index and reverse pulmonary vascular remodeling in PAH animal models. In vitro experiments revealed that its PDE5 inhibitory activity was stronger than sildenafil and tadalafil. Phase-1b clinical trials have been completed with good safety and tolerability [113,114,115], and a phase-2a multicenter, randomized, placebo- and positive-controlled study (NCT04483115) is ongoing to assess its acute hemodynamic effects in PAH patients.

GMA301 is a new monoclonal antibody drug targeting endothelin receptor A that obtained orphan drug qualification from the US FDA in January 2017 and completed its phase-1a clinical trial in Australia. The results showed that GMA301 had good safety and a half-life of 21 days [116,117]. A randomized, double-blind, placebo-controlled, dose-escalation study to assess the safety, efficacy, and pharmacokinetic profile of repeated doses of GMA301 in PAH patients is currently being conducted jointly between China and the United States (NCT04503733).

The PAH vaccine ETRQβ-002 against endothelin receptor A could effectively reduce right ventricular systolic pressure and reverse right ventricular hypertrophy and pulmonary vascular remodeling in PAH animal models with a three-month observation period after vaccine injection [118]. It exhibited long-term therapeutic effects in lowering right ventricular systolic pressure and ameliorating pulmonary vascular remodeling, without immune-mediated damage to hepatic or renal function within 21 weeks [119]; further exploration into its efficacy and safety among PAH patients is required.

## 6. Future Perspective

In the past 30 years, with the in-depth understanding of the pathogenesis of PH, the diagnosis and treatment of PH has made significant progress, from no drugs being available to a wide variety of agents; the quality of life and prognosis of patients have been significantly improved, largely due to early screening, diagnosis, standardized risk stratification, new drug approvals, initial and sequential combined therapy, and dynamic prognostic evaluation. However, the pathogenesis of PH, which involves oxidative stress, immune inflammation, epigenetic modifications, growth factor activation, proliferation and anti-proliferative signal imbalance, hormone signal abnormalities, and so on, is still not fully understood. At present, PAH targeted agents based on three classical signaling pathways significantly improve the prognosis of PAH and CTEPH patients, but the reversal of pulmonary vascular remodeling is limited, and there is still a long way to go to find a cure for the disease. There is still great controversy about when to start combined therapy and how to optimally utilize combined drugs, and the combination strategies adopted by various centers are not the same. While the combined drugs bring benefits, their adverse reactions cannot be ignored. Therefore, the future research direction is not only to explore the best combination strategies of existing targeted drugs, but also to develop novel PAH targeted drugs with the help of multi-omics and other modalities. Omics-based innovative approaches that integrate network medicine and artificial intelligence, such as the PVDOMICS network, may help identify central pathways and discover functional biomarkers and therapeutic targets [120,121].The new fusion protein sotatercep [107], specific potent prostacyclin receptor agonist ralinepag [111], and the ETRQβ-002 vaccine [118] are expected to bring new hope to PAH patients. In recent years, with the continuous implementation of national medical insurance policies, some targeted drugs for the treatment of PAH have entered medical insurance, and the price of PAH targeted drugs has been greatly reduced. With the joint efforts of society, doctors, and patients, PAH patients are expected to witness a brighter tomorrow.

## Figures and Tables

**Figure 1 pharmaceutics-15-01579-f001:**
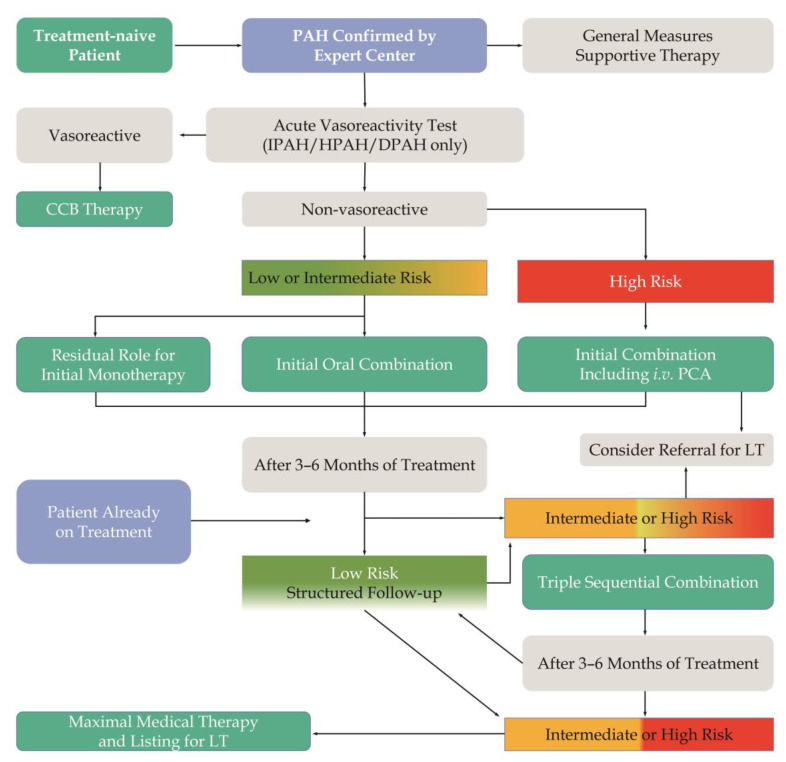
Treatment algorithm of PAH. PAH: pulmonary arterial hypertension; IPAH: idiopathic PAH; HPAH: heritable PAH; DPAH: drug-induced PAH; CCB: calcium channel blocker; PCA: prostacyclin analog; LT: lung transplantation.

**Figure 2 pharmaceutics-15-01579-f002:**
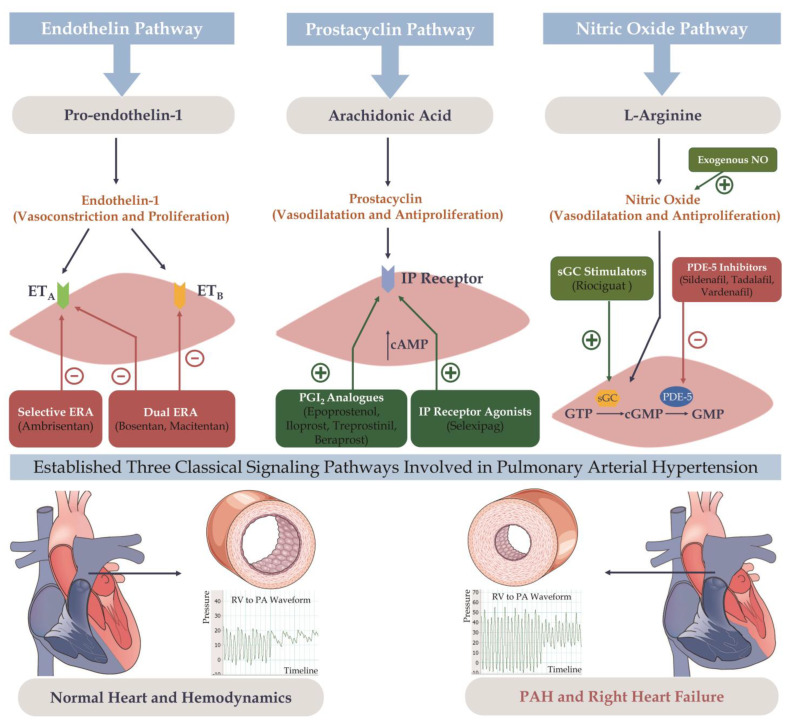
Three classical signaling pathways involved in pulmonary arterial hypertension (PAH) and secondary right heart failure. ET_A_: endothelin receptor A; ET_B_: endothelin receptor B; ERA: endothelin receptor antagonists; cAMP: cyclic adenosine monophosphate; PGI_2_: prostaglandin I2; NO: nitric oxide; sGC: soluble guanylate cyclase; PDE5: phosphodiesterase type 5; GTP: guanosine diphosphate: cGMP: cyclic guanosine monophosphate: GMP: guanosine monophosphate. RV: right ventricle; PA: pulmonary artery.

**Table 1 pharmaceutics-15-01579-t001:** Clinical classification of PH in the 2022 ESC/ERS guidelines for the diagnosis and treatment of PH.

Group 1 Pulmonary arterial hypertension (PAH)	Group 3 PH associated with lung diseases and/or hypoxia
1.1 Idiopathic	3.1 Obstructive lung disease or emphysema
1.1.1 Non-responders at vasoreactivity testing	3.2 Restrictive lung disease
1.1.2 Acute responders at vasoreactivity testing	3.3 Lung disease with mixed restrictive/obstructive pattern
1.2 Heritable	3.4 Hypoventilation syndromes
1.3 Associated with drugs and toxins	3.5 Hypoxia without lung disease
1.4 Associated with:	3.6 Developmental lung disorders
1.4.1 Connective tissue disease	**Group 4 PH associated with pulmonary artery obstructions**
1.4.2 HIV infection	
1.4.3 Portal hypertension	
1.4.4 Congenital heart disease	4.1 Chronic thrombo-embolic PH
1.4.5 Schistosomiasis	4.2 Other pulmonary artery obstructions
1.5 PAH with features of venous/capillary involvement	**Group 5 PH with unclear and/or multifactorial mechanisms**
1.6 Persistent PH of the newborn	
**Group 2 PH associated with left heart disease**	
	5.1 Hematological disorders
2.1 Heart failure:	5.2 Systemic disorders
2.1.1 with preserved ejection fraction	5.3 Metabolic disorders
2.1.2 with reduced or mildly reduced ejection fraction	5.4 Chronic renal failure with or without hemodialysis
2.2 Valvular heart disease	5.5 Pulmonary tumor thrombotic microangiopathy
2.3 Congenital/acquired cardiovascular conditions leading to post-capillary PH	5.6 Fibrosing mediastinitis

PH: pulmonary hypertension; ESC/ERS: European Society of Cardiology/European Respiratory Society; PAH: pulmonary arterial hypertension; HIV: human immunodeficiency virus.

**Table 2 pharmaceutics-15-01579-t002:** Usage and main side effects of current PAH targeted drugs.

PAH Targeted Drug		Usage	Main Side Effects
Endothelin Receptor Antagonists	Bosentan	62.5–125 mg, bid	Elevated transaminase, peripheral edema, anemia
	Ambrisentan	5–10 mg, qd	Headache, peripheral edema, anemia
	Macitentan	10 mg, qd	Anemia
Phosphodiesterase Type 5 Inhibitors	Sildenafil	20 mg, tid	Headache, blushing, increased menstruation
	Tadalafil	20–40 mg, qd	Headache, blushing, muscle soreness
	Vardenafil	5 mg, bid	Headache, blushing, muscle soreness
Soluble Guanylate Cyclase Agonists	Riociguat	1–2.5 mg, tid	Gastrointestinal symptoms, hypotension, hemoptysis
Prostacyclin Analogs	Epoprostenol	Initiated at 2–4 ng/kg/min	Headache, gastrointestinal symptoms, catheter-related infections
	Iloprost	10–20 ug once, 6–9 times/d	Headache, flushing, hypotension
	Treprostinil	Initiated at 1.25 ng/kg/min, iv/sc	Infusion site pain, headache, diarrhea
	Beraprost	20–80 ug, tid-qid	Headache and flushing
Prostacyclin Receptor Agonists	Selexipag	200–1600 μg, bid	Headache, diarrhea, vomiting, jaw pain

**Table 3 pharmaceutics-15-01579-t003:** Investigational drugs for the treatment of PAH.

Type	Drug Name	Mechanism/Feature	Phase	Estimated Enrollment	Recruitment Status	Clinical Trial Number
Endothelin Pathway	GMA301	Humanized monoclonal antibody against ETA	1	48	Recruiting	NCT04503733
Prostacyclin Pathway	Ralinepag	Oral selective prostacyclin IP receptor agonist	2	45	Completed	NCT02279745
			2	61	Completed	NCT02279160
			3	10	Active, not recruiting	NCT04084678
			3	1000	Enrolling by invitation	NCT03683186
			3	1000	Recruiting	NCT03626688
Soluble Guanylate Cyclase Stimulator	BAY1237592	Inhaled soluble guanylate cyclase stimulator	1	38	Completed	NCT03754660
	MK-5475		1	25	Completed	NCT03744637
			2, 3	450	Recruiting	NCT04732221
Phosphodiesterase Inhibitor	Milrinone	Phosphodiesterase type 3 inhibitor	1	40	Completed	NCT04391478
	TPN171H	Phosphodiesterase type 5 inhibitor	2	60	Completed	NCT04483115
	Udenafil		2	63	Completed	NCT01553721
			2, 3	59	Completed	NCT02304198
	Vardenafil RT234		2	14	Completed	NCT05343637
			2b	86	Recruiting	NCT04266197
TGF-β/BMP Signal	Sotatercept	Activin signaling inhibitor rebalancing anti-proliferative and Pro-Proliferative signaling pathways	2	106	Completed	NCT03496207
			2	21	Completed	NCT03738150
			3	324	Completed	NCT04576988
			3	662	Recruiting	NCT04811092
			3	700	Recruiting	NCT04796337
			3	200	Recruiting	NCT04896008
DNA Damage Repair/ Epigenetic Modification	ABI-009	Albumin-bound mTOR inhibitor	1/1b	15	Completed	NCT02587325
	Apabetalone	Selective BET protein inhibitor	Early Phase 1	7	Completed	NCT03655704
			2	72	Not yet recruiting	NCT04915300
	Olaparib	Polymerase inhibitors	1, 2	20	Recruiting	NCT03782818
Hormone Modulators	Anastrozole	Third-generation aromatase inhibitor	2	18	Completed	NCT01545336
			2	84	Completed	NCT03229499
	DHEA	Dehydroepiandrosterone	2	24	Recruiting	NCT03648385
			2	130	Recruiting	NCT03617458
	Fulvestrant	Estrogen receptor antagonist	2	5	Completed	NCT02911844
	rhACE2	Recombinant human ACE2	2	23	Completed	NCT03177603
			NA	1899	Recruiting	NCT01884051
	Rodatristat Ethyl	Tryptophan hydroxylase 1 inhibitor	2	90	Recruiting	NCT04712669
	Spironolactone	Aldosterone receptor antagonist	2	70	Recruiting	NCT01712620
	Tamoxifen	Selective estrogen receptor modulator	2	24	Recruiting	NCT03528902
Inflammation and Immunity	Anakinra	IL-1 receptor antagonist	Ib/II	7	Completed	NCT03057028
	Elafin	Elastase-Specific Protease Inhibitor	1	30	Completed	NCT03522935
	Rituximab	CD20 monoclonal antibody	2	57	Completed	NCT01086540
	Satralizumab	Anti-IL-6 receptor antibody	2	24	Recruiting	NCT05679570
	Sulfasalazine	Disease-modifying anti-rheumatic drug	1, 2	80	Recruiting	NCT04528056
	Tacrolimus	Calcineurin inhibitors	2	23	Completed	NCT01647945
Metabolic Regulator	Benzbromarone	Non-competitive inhibitor of xanthine oxidase	2	10	Completed	NCT02790450
	Dapagliflozin	SGLT2 inhibitor	2	52	Recruiting	NCT05179356
	Empagliflozin		2	8	Not yet recruiting	NCT05493371
	L-Carnitine	Fatty acid metabolism	1	12	Active, not recruiting	NCT04908397
	Metformin	AMPK activator	2	130	Recruiting	NCT03617458
	Trimetazidine	Partial fatty acid oxidation inhibitor	2	25	Completed	NCT02102672
Coagulation and Thrombosis	Ifetroban	Thomboxane A2/prostaglandin H2 receptor antagonist	2	34	Recruiting	NCT02682511
Others	Carvedilol	Non-selective beta-blockers	NA	30	Completed	NCT01586156
	Famotidine	Histamine-2 receptor antagonist	2	80	Recruiting	NCT03554291
	Hymecromone	Inhibitor of hyaluronan synthesis	2	16	Active, not recruiting	NCT05128929
	Imatinib	Tyrosine kinase inhibitors	1	83	Completed	NCT04903730
			2, 3	462	Recruiting	NCT05036135
			2	43	Recruiting	NCT04416750
			3	462	Recruiting	NCT05557942
	LTP001	SMURF1 inhibitor	2	44	Recruiting	NCT05135000
			2	40	Recruiting	NCT05764265
	MN-08	Dual-functional memantine nitrate derivative	1	16	Not yet recruiting	NCT05660863
	Seralutinib	PDGFR kinase inhibitor	2	86	Completed	NCT04456998
			2	100	Recruiting	NCT04816604

PAH: pulmonary arterial hypertension; ETA: endothelin receptor A; TGF-β/BMP: transforming growth factor-beta/bone morphogenetic protein; mTOR: mammalian target of rapamycin; BET: bromodomain and extraterminal domain; DHEA: dehydroepiandrosterone; ACE2: angiotensin-converting enzyme 2; IL-1: interleukin-1; CD20: cluster of differentiate 20; IL-6: interleukin-6; SGLT2: sodium glucose co-transporter 2; AMPK: adenosine monophosphate-activated protein kinase; SMURF1: smad ubiquitination regulatory factor-1; PDGFR: platelet-derived growth factor receptor; NA: not applicable.

## Data Availability

Data are available from the authors upon reasonable request and with the permission of MDPI.

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
