# Peer review of "Medical Management of Pulmonary Arterial Hypertension: Current Approaches and Investigational Drugs"

_pharmaceutics, 2023, doi:10.3390/pharmaceutics15061579_

Round 1

Reviewer 1 Report

The Authors provide a comprehensive review of the current PAH therapeutic agents. The text is well-written and sounds well. 

Comments:

1. The text would benefit of at least 2 figures: the first showing for each drug its molecular target and its benefical effect in PAH and the second summarizing the monotherapy and combination therapy outcomes. This would help the reader to get the home message quickly.  

2. "5. Future Perspective". This section would benefit of a comment on the most innovative research strategy known as Network Medicine and Artificial Intelligence to understand PAH pathobiology and translate biological findings in useful biomarkers or drug targets (mainly network-guided drug repurposing).  (cite:  PMID: 36565787, PMID: 29437835).

Author Response

Reviewer #1: The authors provide a comprehensive review of the current PAH therapeutic agents. The text is well-written and sounds well.

Authors: We would like to thank the reviewer for above kind comments and praise.

Reviewer #1: 1. The text would benefit of at least 2 figures: the first showing for each drug its molecular target and its benefical effect in PAH and the second summarizing the monotherapy and combination therapy outcomes. This would help the reader to get the home message quickly. 

Authors: These are really good advice. As suggested, we added some figures to help the readers get the home message quickly. Figure 1 shows the treatment algorithm of PAH in order to vividly display treatment options based on personalized risk stratifications. Figure 2 shows currently established three classical signaling pathways involved in PAH, in which molecular target of each drug and their beneficial effects in PAH were vividly illustrated.

For the outcomes of monotherapy and combination therapy, as there are currently at least 12 PAH-targeted agents used clinically, and in addition, there are many important clinical trials assessing the efficacy and safety of each agent, as well as a wealth of combination therapy manner, it is impractical for us to present all studies and relevant references in this review, but for the convenience of the readers to obtain relevant information, we cited relevant article (https://doi.org/10.1164/rccm.202109-2079PP, PubMed: 34905704) that well summarize the effects of combination therapy and hope you are satisfied. Thanks once again for your good suggestion.

Reviewer #1: 2. "5. Future Perspective". This section would benefit of a comment on the most innovative research strategy known as Network Medicine and Artificial Intelligence to understand PAH pathobiology and translate biological findings in useful biomarkers or drug targets (mainly network-guided drug repurposing).  (cite:  PMID: 36565787, PMID: 29437835).

Authors: Thanks for your good advice, we added comments and cited the related references as follows “Omics-based innovative approaches that integrate network medicine and artificial intelligence, such as PVDOMICS network, may help identify central pathways and discover functional biomarkers and therapeutic targets”.

Reviewer 2 Report

this is a very elaborated review on the existing and investigational therapies in PAH

generally the available evidence is well summarized especially on the approved therapies and on the used therapeutic regimens

However there are some oversights and clarifications required in order to increase the value of your draft

1. inotropes and vassopresors these therapies are used in all forms of PH when right ventricular failure (usually acute) is present and clinically manifest. so perhaps you should introduce this section with a short paragraph on this issue 

2 . in the same section vassopresors are only vassopresin, dobutamine/dopamine and epinephrine/norepinephrine all the others being definitely only positive inotropes. please correct

3.title says about current perspectives: I would correct it to current approaches. as far as the novel therapeutic targets are concerned, table 3 also contains a great deal of existing therapeutic targets (ie a cytokine, a growth factor or a related pathway) but targeted with new compounds. so that perhaps -Current approaches and investigational drugs- would read better? 

or 

if you want to stick on novel therapeutic targets please make sure that in table 3 those targets are really novel (not considered so far in PAH)

4. Table 3 reads better if potential is replaced with Investigational

5. Table 5 again first column contains both the active substance and the mechanism of action which is also mentioned in the second column. please make sure that you delete the redundant information (perhaps instead of drug  say pharmacological class, and then drug and then in the third column instead of extensively citing NCTs you should mention the latest phase of development (ie the latest phase of clinical trials available - eg phase III, phase Ia or if it is still preclinical). the problem with these ncts  you put in your table is that they usually need to be referenced, and with the huge numbers you have it is rather difficult to have a long and bothersome reference list you can reference the latest clinical phase for example with the most recent NCT and this makes the revision and the peer reviewer's life much easier. Only make sure it is indeed PAH and not CTEPH

Author Response

Reviewer #2: This is a very elaborated review on the existing and investigational therapies in PAH. Generally the available evidence is well summarized especially on the approved therapies and on the used therapeutic regimens.

Authors: We would like to thank the reviewer for above kind comments and praise.

Reviewer #2: However, there are some oversights and clarifications required in order to increase the value of your draft. 1. inotropes and vassopresors these therapies are used in all forms of PH when right ventricular failure (usually acute) is present and clinically manifest. so perhaps you should introduce this section with a short paragraph on this issue.

Authors: Many thanks for pointing out this important aspect, we introduced this section with a short paragraph “In the presence of right ventricular failure manifested with persistent hypotension, inotropes and/or vasopressors are often indicated in all forms of PH to ensure hemodynamic stability and peripheral perfusion”.

Reviewer #2: 2. in the same section vassopresors are only vassopresin, dobutamine/dopamine and epinephrine/norepinephrine all the others being definitely only positive inotropes. please correct.

Authors: This is a very interesting topic. Norepinephrine, an endogenous catecholamine released by postganglionic adrenergic nerves, has potent α receptor activity, which leads to marked peripheral vasoconstriction. It only has modest β1 activity, and therefore has less potent direct inotropic properties. Thus, in the 2018 Cologne Consensus and 6th World Symposium on Pulmonary Hypertension Consensus, norepinephrine and Vasopressin were classified as Vasopressors (https://doi.org/10.1183/13993003.01906-2018; https://doi.org/10.1016/j.ijcard.2018.08.081), Dopamine, Dobutamine and Epinephrine were classified as Inotropes, so we keep the original expression in our manuscript.

Reviewer #2: 3. title says about current perspectives: I would correct it to current approaches. as far as the novel therapeutic targets are concerned, table 3 also contains a great deal of existing therapeutic targets (ie a cytokine, a growth factor or a related pathway) but targeted with new compounds. so that perhaps -Current approaches and investigational drugs- would read better? or if you want to stick on novel therapeutic targets please make sure that in table 3 those targets are really novel (not considered so far in PAH).

Authors: Thanks for your good advice, we modified our title with “Medical Management of Pulmonary Arterial Hypertension: Current Approaches and Investigational Drugs”.

Reviewer #2: 4. Table 3 reads better if potential is replaced with Investigational.

Authors: We revised the title of Table 3 as you suggested “Investigational drugs for the treatment of PAH”, many thanks.

Reviewer #2: 5. Table 5 again first column contains both the active substance and the mechanism of action which is also mentioned in the second column. please make sure that you delete the redundant information (perhaps instead of drug  say pharmacological class, and then drug and then in the third column instead of extensively citing NCTs you should mention the latest phase of development (ie the latest phase of clinical trials available - eg phase III, phase Ia or if it is still preclinical). the problem with these ncts  you put in your table is that they usually need to be referenced, and with the huge numbers you have it is rather difficult to have a long and bothersome reference list you can reference the latest clinical phase for example with the most recent NCT and this makes the revision and the peer reviewer's life much easier. Only make sure it is indeed PAH and not CTEPH.

Authors: Thanks for your kind suggestion. In order to make the table look clear, we added the Phase of development, Estimated Enrollment Number, Recruitment Status, More importantly, we carefully checked each clinical trials, since pulmonary arterial hypertension was focused on, we deleted some studies which included patients with other types of PH (such as pulmonary hypertension due to left heart disease, CTEPH or PH due to lung diseases). It is rather difficult and inappropriate to have a long and bothersome reference list in Tables, so here in our Table 1, we only list the NCT numbers for reference.

Reviewer 3 Report

The paper of Jin et al is very interesting however some issues should be addressed 

1. A short summary of PH pathophysiology should be added

2. Please add the role of levosimendan in the treatment of PH

3.Add an algorithm on the management of PH

Author Response

Reviewer #3: The paper of Jin et al is very interesting however some issues should be addressed.

Authors: We would like to thank the reviewer for above kind comments. Our responses to your comments are as follows.

Reviewer #3: 1. A short summary of PH pathophysiology should be added.

Authors: Thanks for your good advice. We added two short paragraphs on the pathogenesis and pathophysiology of PAH (Page 2-3).

Details are as follows: Different subtypes of PH have divergent pathogenesis and pathophysiological features, which are closely related to the diagnosis and treatment, while this review mainly fo-cuses on PAH. PAH is characterized by small arteriolar intimal hyperplasia, medial hypertrophy, and adventitial fibrosis with variable inflammatory responses and minimal fibrinoid necrosis, and plexiform lesions may develop in the late stages. At present, the pathogenesis of PAH is not fully understood, and abnormal pulmonary vasoconstriction and pulmonary vascular remodeling are two important pathophysiological processes of PAH, while previous studies have suggested that genetic and epigenetic modifications, immune and inflammatory disorders, estrogen dysfunction, oxidative stress-related signaling pathways, and ion channel abnormalities are in-volved in the occurrence and development of PAH[12, 13].

Normal pulmonary vascular beds have strong diastolic and systolic reserve capacity to accommodate the need for increased pulmonary blood flow. However, this ability is compromised in the pulmonary vascular bed among patients with PAH, resulting in increased pulmonary arterial pressure and resistance at rest, especially during exercise. Initially, the right ventricle compensates for the increase in afterload by increasing contractility and ventricular wall thickness, and the right ventricle can maintain normal cardiac output at rest, but cardiac output is somewhat impaired during exercise. As the disease progresses, the right ventricular-pulmonary artery un-couples and cardiac output begins to decline at rest, and as the right heart function deteriorates further, the right ventricle dilates and fails, while the compression of enlarged right ventricle on left ventricle causing left ventricular dysfunction[14].

Reviewer #3: 2. Please add the role of levosimendan in the treatment of PH.

Authors: Actually, in the previous version of our manuscript, we described the role of levosimendan in the treatment of PH with several sentences, in order to emphasize its implication, we updated related contents as follows: Levosimendan is a calcium sensitizer with inotropic, pulmonary vasodilatory, and cardioprotective properties and has been shown to be an effective and safe treatment strategy for PAH patients and PH associated with left heart disease[26]. Levosimendan treatment improved cardiac output, decreased right ventricular afterload, and allevi-ated pulmonary vascular remodeling in SU5416/Hypoxia induced PAH rat models[27]. The most recent systematic review and meta-analysis demonstrated levosimendan improved right ventricular systolic function and reduced pulmonary arterial pressure in patients with cardiac dysfunction[29]; For PAH associated with connective tissue disease (CTD-PAH) patients, levosimendan was effective in improving acute decom-pensated right heart failure, systemic hemodynamics, and in-hospital survival, but no medium- or long-term survival benefit was observed[30]. For patients with low cardiac output, dobutamine and milrinone are the most extensively used inotropes, but levo-simendan appears to be more efficacious than dobutamine in animal models of right heart failure and convincing clinical evidence is still lacking[28].

Reviewer #3: 3. Add an algorithm on the management of PH.

Authors: As suggested, we added Figure 1 for Treatment algorithm of PAH, thanks for your good advice.

Round 2

Reviewer 1 Report

All my concerns have been solved in the appropriate manner.